

# Correlation between obesity and clinicopathological characteristics in patients with papillary thyroid cancer: a study of 1579 cases: a retrospective study

Huijuan Wang[1], Pingping Wang[2], Yu Wu[3], Xiukun Hou[1], Zechun Peng[4], Weiwei Yang[5], Lizhao Guan[6], Linfei Hu[1], Jingtai Zhi[1], Ming Gao[1] and Xiangqian Zheng[1]

[1] Department of Thyroid and Neck Tumor, Tianjin Medical University Cancer Institute and Hospital, National Clinical Research Center for Cancer, Key Laboratory of Cancer Prevention and Therapy, Tianjin, Tianjin's Clinical Research Center for Cancer, Tianjin, China

[2] Department of Thyroid and Breast Surgery, Rizhao Central Hospital, Shandong, China

[3] Department of Head and Neck Surgery, Fujian Cancer Hospital, Fujian Medical University Cancer Hospital, Fujian, China

[4] Department of Urology Surgery, AreaII, The Second Affiliated Hospital of Hainan Medical University, Hainan, China

[5] Department of Otolaryngology-Head and Neck Surgery, Tianjin First Center Hospital, Tianjin, China

[6] Department of Biochemistry and Molecular Biology, School of Basic Medical Sciences, Tianjin Medical University, Tianjin, China

Corresponding author
Xiangqian Zheng,
xzheng05@tmu.edu.cn,
xiangqian_zheng@163.com

## ABSTRACT

**Objective**. To explore the relationship between body mass index (BMI) and clinicopathological characteristics in patients with papillary thyroid carcinoma (PTC).

**Methods**. The clinical data of 1,579 patients with PTC, admitted to our hospital from May 2016 to March 2017, were retrospectively analyzed. According to the different BMI of patients, it can be divided into underweight recombination (BMI $< 18.5$ kg/m), normal body recombination ($18.5 \leq$ BMI $< 24.0$ kg/m$^2$), overweight recombination ($24.0 \leq$ BMI $< 28.0$ kg/m$^2$) and obesity group (BMI $\geq 28.0$ kg/m$^2$). The clinicopathological characteristics of PTC in patients with different BMIs group were compared.

**Results**. In our study, the risk for extrathyroidal extension (ETE), advanced T stage (T III/IV), and advanced tumor-node-metastasis stage (TNM III/IV) in the overweight group were higher, with OR (odds ratio) $= 1.99(1.41–2.81)$, OR $= 2.01(1.43–2.84)$, OR $= 2.94(1.42–6.07)$, respectively, relative to the normal weight group. The risk for ETE and T III/IV stage in the obese group were higher, with OR $= 1.82(1.23–2.71)$ and OR $= 1.82(1.23–2.70)$, respectively, relative to the normal weight group.

**Conclusion**. BMI is associated with the invasiveness of PTC. There is a higher risk for ETE and TNM III/IV stage among patients with PTC in the overweight group and for ETE among patients with PTC in the obese group.

## INTRODUCTION

The incidence of thyroid cancer has been increasing in recent years worldwide. Thyroid cancer in women has become the fifth most common malignant tumor in the United States (*Siegel, Miller & Jemal, 2018*). Thyroid cancer has become the most common tumor in women in South Korea (*McGuire, 2016*). Thyroid screening and over-diagnosis do not explain the significant increase in the incidence of primary tumors $\geq$ 4 cm and the incidence of distant metastasis. Although the rate of thyroid cancer detection has improved, the survival rate has not increased. This indicates that it is necessary to further explore the causes of the increase in the incidence of thyroid cancer which cannot simply be explained by the increase in detection rates. It is also necessary to study this problem from the perspective of factors such as environmental factors and molecular mechanisms (*Chen, Jemal & Ward, 2009*; *Enewold et al., 2009*). The real cause of the increase in the incidence of thyroid cancer has not yet been determined; however, environmental factors or lifestyle may contribute to this increase. Several epidemiological studies have confirmed that obesity is positively correlated with the increased risk of thyroid cancer (*Pappa & Alevizaki, 2014*; *Schmid et al., 2015*; *Patel et al., 2015*; *Yin et al., 2018*). However, the correlation between obesity and the invasive clinicopathological features of thyroid cancer remains controversial (*Kwon et al., 2015*; *Paes et al., 2010*; *Grani et al., 2019*; *Lee et al., 2015*). In this study, the Chinese body mass index (BMI) classification criteria were used to explore whether the clinicopathological characteristics of PTC are different among patients with different BMIs.

## MATERIALS & METHODS

### Patients

A total of 1,702 patients with PTC (including thyroid micropapillary carcinoma, PTMC) who received surgical treatment in Tianjin Medical University Cancer Institute and Hospital from May 2016 to March 2017 were considered. After excluding patients with histories of thyroid surgery, antithyroid drug consumption, and thyroxine administration before surgery, 1,579 subjects were eligible for analysis in this study. Each participant signed an informed consent form, which was uploaded in supplementary materials. This study was approved by the Ethics Committee of the Tianjin Medical University Cancer Institute and Hospital. Ethics Committee reference number is Ek2018117.

### Methods

We performed a retrospective analysis of the patient's gender, age, serum thyroid stimulating hormone (TSH) levels, combined with postoperative pathological features, including tumor size (maximum diameter of the tumor), lymph node metastasis, multifocality, and the extrathyroidal extension (ETE) and TNM stage based on the eighth edition of the Union for International Cancer Control (UICC)/American Joint Committee on Cancer (AJCC). We reviewed the height and the weight of the patient during admission, calculated BMI according to the Chinese obesity classification standard (BMI < 18.5 kg/m$^2$, underweight; 18.5 $\leq$ BMI < 24.0 kg/m$^2$, normal weight; 24.0 $\leq$

BMI$< 28.0$ kg/m$^2$, overweight; and BMI $\geq 28.0$ kg/m$^2$, obese) (*Qian, Li & Ren, 2017*). Subsequently, the pathological characteristics including multifocality, tumor size, ETE, lymph node metastasis, T stage, TNM stage of each group were compared.

### Statistical analysis

Logistic regression analysis was used to analyze the relationship between BMI and the clinical pathological features of thyroid cancer. The odds ratio (OR) and 95% confidence interval were used. The adverse clinicopathological features analyzed included multifocality (number of lesions $\geq 2$), tumor size $\geq 1$ cm, ETE, lymph node metastasis, high T stage (stage III + IV), and high TNM stage (stage III + IV). Logistic regression (adjusting for age, gender and TSH) was used to analyze the relationship between BMI and the adverse clinicopathological features of PTC. Similarly, logistic regression analysis (adjusting for age and TSH) was used to analyze the relationship between BMI and adverse clinicopathological features of PTC in men and women. For those older than $\geq 55$ years and $<55$ years, logistic regression analysis (adjusting for gender and TSH) of the relationship between BMI and adverse clinicopathological features of PTC was performed.

The Chi-square test was used to analyze whether there were differences in gender, age, level of TSH, number of tumors, tumor size, ETE, lymph node metastasis, T stage, and TNM stage among different BMI groups.

Statistical analysis was performed using SAS V9.3 software (Cary, North Carolina, USA) with a statistical significance noted at $P<0.05$.

## RESULTS

Basic clinical biological characteristics of 346 males and 1,233 females were recorded. The age ranged from 18 to 76 years, with an average age of ($45.98 \pm 10.93$) years, a median age of 46 years, 1,129 patients (71.5%) aged $<55$ years, and 450 (28.5%) aged $\geq 55$ years. BMI ranged from 16.00 to 48.33 kg/m$^2$ with mean BMI $25.52 \pm 3.79$ kg/m$^2$. A total of 704 patients (44.6%) had lymphatic metastasis, 228 (14.4%) had ETE and 565 (35.5%) had multifocal tumors. With regards to the T stage, 1,322 (83.7%) patients were in the T1 stage and 257 patients (16.3%) were in the T3/4 stage. With regards to the TNM stage, 1,515 (95.9%) patients were in stage I and II, and 64 patients (4.1%) were in stage III and IV (Table 1).

There are differences in the distribution of gender ($\chi^2 = 80.28$, $P<0.0001$) and age ($\chi^2 = 27.05$, $P<0.0001$) between different BMI groups. BMI is associated with invasion of the envelope ($\chi^2 = 22.25$, $P<0.0001$), T stage ($\chi^2 = 22.81$, $P<0.0001$), and TNM stage ($P = 0.0002$) in the pathological features of the tumor (Table 2).

We further explored the risk of more aggressive clinicopathological features according to BMI (Table 3). Multiple logistic regression results display that patients who were overweight had a significantly greater risk of ETE (OR $= 1.99[1.41–2.81]$, $P<0.0001$), high T stage (OR $= 2.01[1.43–2.84]$, $P<0.0001$), and TNM III/IV stage (OR $= 2.94[1.42–6.07]$, $P = 0.003$) than patients with a normal weight. Subjects in the obese group also had a greater risk of ETE (OR $= 1.82[1.23–2.71]$, $P = 0.002$) and high T stage (OR $= 1.82[1.23–2.70]$,

**Table 1 Clinicopathological characteristics of 1,579 patients with papillary thyroid carcinoma.**

| Clinicopathological characteristics $n = 1{,}579$ | |
|---|---|
| Gender | |
| Female | 1,233(78.1%) |
| Male | 346 (21.9%) |
| age | 45.98 ± 10.93 |
| < 55 | 1,129(71.5%) |
| ≥55 | 450 (28.5%) |
| Tumor size | |
| <1 cm | 906 (57.4%) |
| ≥1 cm | 673 (42.6%) |
| Extrathyroidal invasion | 228 (14.4%) |
| multifocality | 565 (35.5%) |
| T staging | |
| T 1 | 1,322(83.7%) |
| T2 | 28 (1.8%) |
| T3 | 153 (9.7%) |
| T4 | 76 (4.8%) |
| N staging | |
| N0 | 875(55.4%) |
| N1a | 441(27.9%) |
| N1b | 263(16.7%) |
| TNM staging | |
| I/II | 1,515(95.9%) |
| III/IV | 64 (4.1%) |

$P = 0.003$) than normal weight subjects. Whether in the overweight or obese group, BMI has no correlation with lymph node metastasis.

Among female patients, compared to the normal weight group, the overweight group had a greater risk of ETE (OR = 2.10[1.43–3.08], $P = 0.0002$), high T stage (OR = 2.10[1.43–3.08], $P = 0.0002$), and TNM III/IV tumors (OR = 2.86[1.18–6.94] , $P = 0.02$); the obese group had a greater risk of ETE (OR = 2.45[1.58–3.82], $P<0.000$), high T stage (OR = 2.45[1.58–3.82], $P<0.000$), and TNM III/IV tumors (OR = 3.99[1.55–10.28], $P = 0.0004$) (Table 4). In male patients, no significant differences were observed (S1).

When the patient's age was ≥ 55 years, ETE, high T stage, and TNM III/IV tumors were more common in the overweight group than in the normal weight group, with ORs = 2.19(1.22–3.89), $P = 0.009$, ORs = 2.18(1.22–3.89), $P = 0.008$, and ORs = 2.42(1.15–5.13), $P = 0.02$, respectively. ETE (OR = 2.03[1.06–3.89], $P = 0.03$) and high T stage (OR = 2.03[1.06–3.89], $P = 0.03$) were each more frequent in the obese group than in the normal weight group (Table 5).

When the patient's age was < 55 years, ETE and high T stage tumors were more common in the overweight group than in the normal weight group, with ORs = 1.77(1.14–2.74), $P = 0.01$, ORs = 1.80 (1.16–2.78), $P = 0.008$ respectively. ETE (OR = 1.70 [1.02–2.83],

**Table 2  Demographic and clinico-pathological characteristics of patients with different BMI.**

| characteristic | BMI < 18.5 N (%) | 18.5 ≤ BMI < 24 N (%) | 24 ≤ BMI < 28 N (%) | BMI ≥ 28 N (%) | $\chi^2$ | P |
|---|---|---|---|---|---|---|
| gender | | | | | | |
| male | 0(0.00) | 78(12.44) | 141(24.23) | 127(35.98) | 80.28 | <0.0001 |
| female | 17(100.00) | 549(87.56) | 441(75.77) | 226(64.02) | | |
| age | | | | | | |
| <55 | 14(82.35) | 494(78.79) | 378(64.95) | 242(68.75) | 27.05 | <0.0001 |
| ≥55 | 3(17.65) | 133(21.21) | 204(35.05) | 110(31.25) | | |
| TSH | | | | | | |
| normal | 16(94.12) | 590(94.10) | 552(94.85) | 327(92.63) | 1.92 | 0.5884 |
| abnormal | 1(5.88) | 37(5.90) | 30(5.15) | 26(7.37) | | |
| Number of tumors | | | | | | |
| 1 | 12(70.59) | 415(66.19) | 373(64.09) | 218(61.76) | 2.26 | 0.5207 |
| ≥2 | 5(29.41) | 212(33.81) | 209(35.91) | 135(38.24) | | |
| Tumor size | | | | | | |
| <1 | 8(47.06) | 374(59.65) | 326(56.01) | 198(56.09) | 2.74 | 0.4327 |
| ≥1 | 9(52.94) | 253(40.35) | 256(43.99) | 155(43.91) | | |
| Extrathyroidal extension | | | | | | |
| absent | 16(94.12) | 567(90.43) | 475(81.62) | 293(83.00) | 22.25 | <0.0001 |
| present | 1(5.88) | 60(9.57) | 107(18.38) | 60(17.00) | | |
| lymph node metastasis | | | | | | |
| absent | 8(47.06) | 348(55.50) | 332(57.04) | 187(52.97) | 1.96 | 0.5810 |
| present | 9(52.94) | 279(44.50) | 250(42.96) | 166(47.03) | | |
| T staging | | | | | | |
| I + II | 16(94.12) | 567(90.43) | 474(81.44) | 293(83.00) | 22.81 | <0.0001 |
| III + IV | 1(5.88) | 60(9.57) | 108(18.56) | 60(17.00) | | |
| TNM staging | | | | | | |
| I + II | 17(100.00) | 617(98.41) | 545(93.64) | 336(95.18) | — | 0.0002[*] |
| III + IV | 0(0.00) | 10(1.59) | 37(6.36) | 17(4.82) | | |

Notes.
[*]Fisher's exact test was performed because one expected frequency less than 1.

$P = 0.04$), high T stage (OR $= 1.68[1.01–2.80]$, $P = 0.04$) and multifocality (OR $= 1.50$ $[1.08–2.09]$, $P = 0.02$) were each more frequent in the obese group than in the normal weight group (Table 6).

# DISCUSSION

Thyroid cancer is the most common malignant tumor in the endocrine system. Its incidence has increased year by year in the past 20 years. In 2012, the number of new cases of thyroid cancer in China accounted for 15.6% of the global number of new cases, and the number of deaths accounted for 13.8%2. PTC is the most common histological type of thyroid cancer, accounting for about 80% of its incidence (*Ahmad et al., 2018*). In recent decades, advances in thyroid ultrasonography, increased use of fine needle biopsy, and occasional findings from other neck imaging studies have been made; however, these do not fully explain the

**Table 3  Logistic regression of BMI level on different adverse clinico-pathological characteristics.**

|  | BMI < 18.5<br>N = 17 | 18.5 ≤ BMI < 24<br>N = 627 | 24 ≤ BMI < 28<br>N = 582 | BMI ≥ 28<br>N = 353 |
|---|---|---|---|---|
| **Multifocality** | | | | |
| OR (95%CI) | 0.80(0.28,2.31) | Reference | 1.12(0.88,1.43) | 1.26(0.95,1.66) |
| P | 0.68 | | 0.36 | 0.10 |
| **tumor size ≥ 1 cm** | | | | |
| OR (95%CI) | 1.69(0.64,4.46) | Reference | 1.13(0.89,1.43) | 1.12(0.84,1.45) |
| P | 0.28 | | 0.29 | 0.47 |
| **Extrathyroidal extension** | | | | |
| OR (95%CI) | 0.60(0.08,4.65) | Reference | 1.99(1.41,2.81) | 1.82(1.23,2.71) |
| P | 0.63 | | <0.0001 | 0.002 |
| **lymph node metastasis** | | | | |
| OR (95%CI) | 1.47(0.56,3.87) | Reference | 0.92(0.73,1.16) | 1.02(0.78,1.34) |
| P | 0.43 | | 0.48 | 0.88 |
| **T staging(stage III + IV)** | | | | |
| OR (95%CI) | 0.61(0.08,4.66) | Reference | 2.01(1.43,2.84) | 1.82(1.23,2.70) |
| P | 0.63 | | <0.0001 | 0.003 |
| **TNM staging (stage III + IV)** | | | | |
| OR (95%CI) | — | Reference | 2.94(1.42,6.07) | 2.23(0.99,5.05) |
| P | — | | 0.003 | 0.05 |

**Table 4  Logistic regression of BMI level on different adverse clinico-pathological characteristics (female).**

|  | BMI < 18.5<br>N = 17 | 18.5 ≤ BMI < 24<br>N = 549 | 24 ≤ BMI < 28<br>N = 441 | BMI ≥ 28<br>N = 226 |
|---|---|---|---|---|
| **Multifocality** | | | | |
| OR (95%CI) | 0.80(0.28,2.30) | Reference | 1.07(0.82,1.39) | 1.34(0.97,1.85) |
| P | 0.68 | | 0.64 | 0.08 |
| **tumor size ≥1 cm** | | | | |
| OR (95%CI) | 1.68(0.64,4.44) | Reference | 1.14(0.89,1.48) | 1.07(0.78,1.47) |
| P | 0.29 | | 0.30 | 0.68 |
| **Extrathyroidal extension** | | | | |
| OR (95%CI) | 0.64(0.08,4.95) | Reference | 2.10(1.43,3.08) | 2.45(1.58,3.82) |
| P | 0.67 | | 0.0002 | <0.000 |
| **lymph node metastasis** | | | | |
| OR (95%CI) | 1.53(0.58,4.03) | Reference | 0.94(0.73,1.22) | 1.19(0.88,1.64) |
| P | 0.39 | | 0.66 | 0.27 |
| **T staging (stage III + IV)** | | | | |
| OR (95%CI) | 0.64(0.08,4.95) | Reference | 2.10(1.43,3.08) | 2.45(1.58,3.82) |
| P | 0.67 | | 0.0002 | <0.000 |
| **TNM staging (stage III + IV)** | | | | |
| OR (95%CI) | — | Reference | 2.86(1.18,6.94) | 3.99(1.55,10.28) |
| P | — | | 0.02 | 0.0004 |

**Table 5  Logistic regression of BMI level on different adverse clinico-pathological characteristics (age ≥55).**

| | BMI < 18.5 N = 3 | 18.5 ≤ BMI < 24 N = 133 | 24 ≤ BMI < 28 N = 204 | BMI ≥ 28 N = 110 |
|---|---|---|---|---|
| **Multifocality** | | | | |
| OR (95%CI) | — | Reference | 1.02(0.65,1.60) | 0.76(0.45,1.31) |
| P | — | | 0.95 | 0.32 |
| **tumor size ≥1 cm** | | | | |
| OR (95%CI) | — | Reference | 1.38(0.88,2.15) | 1.28(0.77,2.14) |
| P | — | | 0.16 | 0.34 |
| **Extrathyroidal extension** | | | | |
| OR (95%CI) | — | Reference | 2.19(1.22,3.89) | 2.03(1.06,3.89) |
| P | — | | 0.009 | 0.03 |
| **lymph node metastasis** | | | | |
| OR (95%CI) | — | Reference | 1.07(0.68,1.67) | 1.18(0.71,1.98) |
| P | — | | 0.78 | 0.52 |
| **T staging (stage III + IV)** | | | | |
| OR (95%CI) | — | Reference | 2.18(1.22,3.89) | 2.03(1.06,3.89) |
| P | — | | 0.008 | 0.03 |
| **TNM staging (stage III + IV)** | | | | |
| OR (95%CI) | — | Reference | 2.42(1.15,5.13) | 2.01(0.87,4.67) |
| P | — | | 0.02 | 0.10 |

**Notes.**
—The number of people in this group is too small to calculate the correlation.

**Table 6  Logistic regression of BMI level on different adverse clinico-pathological characteristics (age <55).**

| | BMI < 18.5 N = 14 | 18.5 ≤ BMI < 24 N = 494 | 24 ≤ BMI < 28 N = 378 | BMI ≥ 28 N = 243 |
|---|---|---|---|---|
| **Multifocality** | | | | |
| OR (95%CI) | 1.13(0.37,3.44) | Reference | 1.02(0.65,1.60) | 0.76(0.45,1.31) |
| P | 0.82 | | 0.95 | 0.32 |
| **tumor size ≥1 cm** | | | | |
| OR (95%CI) | 2.68(0.88,8.16) | Reference | 1.38(0.88,2.15) | 1.28(0.77,2.14) |
| P | 0.08 | | 0.16 | 0.34 |
| **Extrathyroidal extension** | | | | |
| OR (95%CI) | 0.83(0.11,6.49) | Reference | 2.19(1.22,3.89) | 2.03(1.06,3.89) |
| P | 0.86 | | 0.009 | 0.03 |
| **lymph node metastasis** | | | | |
| OR (95%CI) | 1.69(0.58,4.95) | Reference | 1.07(0.68,1.67) | 1.18(0.71,1.98) |
| P | 0.34 | | 0.78 | 0.52 |
| **T staging (stage III + IV)** | | | | |
| OR (95%CI) | 0.83(0.11,6.52) | Reference | 2.18(1.22,3.89) | 2.03(1.06,3.89) |
| P | 0.86 | | 0.008 | 0.03 |

increasing incidence of PTC, including stage III and IV PTC. Some scholars speculate that this incidence may be affected by other factors such as the environment and lifestyle 3,4.At the same time, several epidemiological studies on obesity and cancer have found that the risks of endometrial, colorectal, breast, thyroid, and prostate cancer are closely related to BMI, and the risk of PTC is positively correlated with BMI (*Lauby-Secretan et al., 2016*; *Ma et al., 2015*). It is concerning that with the urbanization of China, the number of overweight and obese patients has become high, and the Chinese population is no longer a population with a low average BMI. According to statistics, overweight and obese people account for close to 29.2% of the total population of China (*Gordon-Larsen, Wang & Popkin, 2014*). In this study, the 17 underweight patients accounted for only 1% of the patients enrolled, while those who were overweight and obese accounted for 59.2%. The epidemiology of obesity and PTC is shows significant time-trend correlations, suggesting that obesity acts as a risk factor for the occurrence and development of PTC (*Pappa & Alevizaki, 2014*).

At present, the relationship between obesity and the pathological features of PTC remains controversial. *Kim et al. (2016)* found that the risk of ETE among patients with PTC increases with the increase in BMI, and is closely related to the multifocality of the tumor. Another study showed that elevated BMI is associated with tumor size and TNM staging (*Dieringer et al., 2015*). Our study used the Chinese BMI standard and the TNM staging of the eighth edition of AJCC for all patients. Based on multiple logistic regression, the results showed that the proportion of TNM III/IV tumors (OR = 2.86[1.18–6.94], $P = 0.02$) and the risk of ETE (OR = 1.99[1.41–2.81], $P < 0.0001$) increased significantly in overweight group, while tumor size, lymph node metastasis, and multifocal tumors were not significantly associated with BMI; the risk of ETE (OR = 1.82[1.23–2.71], $P = 0.002$) in the obese group increased with BMI. *Kim et al. (2015)* found that BMI is associated with tumor invasion, lymphatic invasion, lymph node metastasis, and tumor multifocality, in patients with PTC. In contrast, some studies suggest that there is no significant correlation between obesity, and clinical pathological features and the recurrence of PTC (*Kwon et al., 2015*; *Grani et al., 2019*). It is worth noting that clinical BMI has certain limitations as the sole criterion for assessing obesity, especially when it reflects the lack of specificity in centripetal obesity (*Rosen & Spiegelman, 2014*). This may be an important reason for the difference in the conclusions of the above studies. We look forward to establishing a more comprehensive obesity evaluation index system, including BMI and abdominal circumference index, in future research.

At present, molecular mechanisms related to obesity and tumors indicate that obesity can promote tumor invasion and metastasis through a variety of obesity-related factors and metabolic pathways (*Marcello et al., 2014*; *Avgerinos et al., 2019*). Adiponectin can reduce the expression of vascular endothelial growth factor (VEGF) and B-cell lymphoma factor-2 (Bcl-2), increase the activity of tumor suppressors such as P53, and inhibit tumor growth and survival. Obesity causes a decrease in adiponectin, and the loss of its receptor expression may be an important mechanism for promoting the progression of PTC. Leptin can increase the expression of VEGF, interleukin-6 (IL-6), and tumor necrosis factor-α (TNF-α) to promote progression and metastasis of thyroid cancer (*Vansaun, 2013*). Overexpression of leptin and its receptors is significantly associated with the

aggressiveness of thyroid cancer (*Fan & Li, 2015*). *Park et al., (2018)* found that a high-fat diet induced more aggressive pathological changes, which were mediated by increased activation of the Janus kinase 2-signaling transducer, activation of the transcription 3 (STAT3) signaling pathway, and induction of STAT3 target gene expression. The discovery of these mechanisms not only reveals the potential molecular basis of obesity as a risk factor in the development and progression of thyroid cancer, but also provides a new therapeutic direction for the future.

## CONCLUSIONS

In summary, obesity is closely related to the risk of PTC and the invasiveness of tumors. Controlling body weight through regular exercise and a reasonable diet and reducing obesity should be important prevention and treatment methods for patients with papillary thyroid cancer and high-risk groups.

### Funding
This work was supported by grants from the National Natural Science Foundation of China (Grant Nos. 81872169, 81702629), the Tianjin key research and development program science and technology support key projects (Grant No. 17YFZCSY00690), and the Tianjin Municipal Science and technology project (Grant No. 19JCYBJC27400). There was no additional external funding received for this study. The funders had no role in study design, data collection and analysis, decision to publish, or preparation of the manuscript.

### Grant Disclosures
The following grant information was disclosed by the authors:
National Natural Science Foundation of China: 81872169, 81702629.
Tianjin key research and development program science and technology support key projects: 17YFZCSY00690.
Tianjin Municipal Science and technology project: 19JCYBJC27400.

### Competing Interests
The authors declare there are no competing interests.

### Author Contributions
- Huijuan Wang, Yu Wu, Xiukun Hou and Zechun Peng performed the experiments, prepared figures and/or tables, and approved the final draft.
- Pingping Wang, Linfei Hu and Jingtai Zhi analyzed the data, authored or reviewed drafts of the paper, and approved the final draft.
- Weiwei Yang and Lizhao Guan analyzed the data, prepared figures and/or tables, and approved the final draft.
- Ming Gao and Xiangqian Zheng conceived and designed the experiments, authored or reviewed drafts of the paper, and approved the final draft.

## Human Ethics

The following information was supplied relating to ethical approvals (i.e., approving body and any reference numbers):

This study was approved by the Ethics Committee of the Tianjin Medical University Cancer Institute and Hospital (Ek2018117).

## Data Availability

Anonymized raw data are available as a Supplemental File.

## Supplemental Information

Supplemental information for this article can be found online at http://dx.doi.org/10.7717/peerj.9675#supplemental-information.

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
