# Peer review of "Correlation between obesity and clinicopathological characteristics in patients with papillary thyroid cancer: a study of 1579 cases: a retrospective study"

_PeerJ, doi:10.7717/peerj.9675_

## Round 0.1 · original submission · Major Revisions

I recommend you carefully revise this manuscript based on the comments of the three reviewers.

Reviewer 1 ·

Basic reporting

This article is written well for the most part. Paragraphs and sentences needs to be split up as noted below. It has enough references.
Tables are fine but might be a bit too many. However, I think some additional analyses should be done.

Experimental design

This was done well.

Validity of the findings

Mostly ok but there are some issues with confidence intervals and data as noted below.

Additional comments

This article was written by a group from China about association of BMI with thyroid cancer aggressiveness. This is one of many studies of its kind but may be one of the largest single institution study. Usually the benefit of single institution is that details that would have otherwise not been seen on large national databases can be seen. I believe the study was done well but could use some improvements.

MAJOR revisions:
1. Methods, line 77: How was TNM stage calculated? On presentation? I assume this is the case, but this should be mentioned. Also how did you determine TNM stage IV? via RAI scan 6 weeks postop?
2. Methods line 82: Could the authors do analyses and split the obese from the morbidly obese categories? What are the BMI groups for those (since they differ in China)
3. Line 102—why is menopause included in this article? What is the evidence on how this affects thyroid cancer other than age?
4. Lines 110-111, the details for this should be added (not just “there is a difference”)
5. Line 116: did you split this analysis by age? I saw the atble with over 55, but truly the TNM/AJCC stages being used for all patients is confusing to assess since they differ by age. SO both above and below age 55 tables should be added. Also, can you provide anaylsis also based on aggressive features rather than AJCC? i.e. nodal disease/metastatic disease, etc? that way age isn't as much of a factor?
6. Line 134—I am not sure this is accurate data as written. These authors added both new cases and deaths—but it needs to be clearer. Does this mean that of everyone who died from thyroid cancer, 14% of the deaths occurred in China?
7. The whole page of lines 147-161—is way too long. It needs to be split up into paragraphs. Very hard to read
8. Lines 152-156—which study is “this” referring to? The one I am reviewing or the one you have referenced?
9. Table 3—Lymph node metastases should be added to the text –this is important
10. Table 6—the confidence intervals are not normal. Why is this and please correct. I think some data was included that has skewed this and should be removed.
MINOR revisions:
1. The authors’ degrees are not written so it is difficult for me to assess their level of competency. Please add those to the manuscript
2. In the Abstract—the author’s should include the definitions of obesity that they used since it is different in China than other parts of the world. They should also detail all the different BMI groups here
3. T stage should be listed in numeric form (3/4) and TNM should be listed in roman numerals (III/IV)—please change throughout manuscript
4. The sentence in line 105 is too long—needs to be split up

Reviewer 2 ·

Basic reporting

Overall well written with no particular concerns about English language grammar or spelling.

Few small issues:

1) The discussion is one long paragraph. It would be much easier to read broken up into several smaller paragraphs.

2) In line 60, "body mass index (BMI)" should replace BMI, as the abbreviation BMI has not yet been explained in the text (only the abstract)

Experimental design

Few issues:

1) Why did the authors collect, as per the materials and methods, data about cholesterol, menopause and TSH? It is never justified about why these parameters may be clinically relevant in thyroid cancer. Furthermore, these data are not taken into account in the multivariate logistic regression analysis either.

2) In line 75 it is mentioned that the authors collected "other clinical data". What was this, and why is it relevant?

3) in line 82, it is mentioned "Subsequently, the pathological characteristics of each group were compared". What pathological characteristics does this isolated sentence refer to?

4) The section "Statistical analysis" seems to misrepresent the role of logistic regression. It is stated that "after adjusting for age and gender, logistic regression was used to analyse the relationship between BMI...". However, from my understanding, multivariate logistic regression is the process by which age and gender are adjusted for. It is not performed "after" adjusting for age and gender. The similar criticism applies to the subsequent sentences up to line 94.

5) Why did the authors not choose to present data about the relationship between age and gender and poor thyroid cancer outcomes in multivariate logistic regression? Is there an independent correlation between age and these outcomes, and gender and these outcomes?

Validity of the findings

Several issues:
1) There are no p- values provided in Tables 3-5, nor in the results section discussing the logistic regression. THis is particularly relevant looking at the frequent odds ratios of <0.01 among the underweight group, which is likely not statistically significant due to the very low numbers in this group and therefore not of relevance.

2) It is unclear why the authors performed a separate analysis for the subgroup of those with PTMC. This was never referenced in the materials and methods section. It was never explained in the results section how many patients had PTMC either.

3) from line 113 on to line 130, it needs to be clearly stated that these results are from multivariate logistic regression controlling for age, gender etc. It is inadequate to simply provide odds ratios here.

Reviewer 3 ·

Basic reporting

Tables and figures are nicely constructed, however P value in tables are missing. Try to use short sentences. Please check the format and grammatical mistakes.

Experimental design

Written very well

Validity of the findings

Need more validation of the output. It is fine, but data needs proper handling.

Additional comments

Line 75: What is other clinical data? How you distinguish TNM stages?
Line 79: What is BMI?
Line 74 : I do not get involvement of menopause in this article? is there any effect on the patient results?
Line 110: What is the difference used in this case. please explain it.
Line 133: Seems data is not constructed accurately, lack of proper handling of the data as i contain both deaths and the new cases of thyroid cancer.
Discussion: Please write it in a paragraph. Please add your result based on regression.

---

## Round 0.2 · accepted · Accept

Post clearance from reviewer's comments, I am informing you the acceptance of your article.

Reviewer 1 ·

Basic reporting

This is clearly written. There are some grammatical errors which the editor can proofread.

Experimental design

Well done

Validity of the findings

Well done

Additional comments

I thank the author for changing the article to include consideration of my comments. I have no further comments. This is a strong article.

Reviewer 2 ·

Basic reporting

My previous concerns have been addressed well.

Experimental design

My previous concerns have been addressed well.

Validity of the findings

I understand that the authors want to purely focus on the independent effects of BMI on PTC outcomes, instead of the independent effects of age and gender.

In this case, I think it is important that the authors reiterate in Tables 3-6 that these analyses have been performed with multivariate logistic regression analysis controlling for age/gender/TSH levels where appropriate. Otherwise it looks like these tables are just performing univariate logistic regression analysis of BMI.

Similarly, it is important that from line 116 to 137, the authors continually reiterate that their findings are independent of age/gender/TSH where appropriate via multivariate analysis. Otherwise it looks like this text is simply describing the results of univariate logistic regression analysis of BMI within the cohort and subcohorts.